# Automatic Annotation of Hyperspectral Images and Spectral Signal Classification of People and Vehicles in Areas of Dense Vegetation with Deep Learning

**Adam Papp, Julian Pegoraro, Daniel Bauer, Philip Taupe \*, Christoph Wiesmeyr and Andreas Kriechbaum-Zabini**

AIT Austrian Institute of Technology GmbH, Giefinggasse 4, 1210 Vienna, Austria; adam.papp@ait.ac.at (A.P.); julian.pegoraro@ait.ac.at (J.P.); daniel.bauer@ait.ac.at (D.B.); philip.taupe@ait.ac.at (P.T.); christoph.wiesmeyr@ait.ac.at (C.W.); andreas.kriechbaum-zabini@ait.ac.at (A.K.-Z.)

\* Correspondence: philip.taupe@ait.ac.at

**Abstract:** Despite recent advances in image and video processing, the detection of people or cars in areas of dense vegetation is still challenging due to landscape, illumination changes and strong occlusion. In this paper, we address this problem with the use of a hyperspectral camera—installed on the ground or possibly a drone—and detection based on spectral signatures. We introduce a novel automatic method for annotating spectral signatures based on a combination of state-of-the-art deep learning methods. After we collected millions of samples with our method, we used a deep learning approach to train a classifier to detect people and cars. Our results show that, based only on spectral signature classification, we can achieve an Matthews Correlation Coefficient of 0.83. We evaluate our classification method in areas with varying vegetation and discuss the limitations and constraints that the current hyperspectral imaging technology has. We conclude that spectral signature classification is possible with high accuracy in uncontrolled outdoor environments. Nevertheless, even with state-of-the-art compact passive hyperspectral imaging technology, high dynamic range of illumination and relatively low image resolution continue to pose major challenges when developing object detection algorithms for areas of dense vegetation.

**Keywords:** hyperspectral imaging; deep learning; computer vision; automatic annotation

## 1. Introduction

Multispectral and hyperspectral imaging allows more detailed observation of the spectral signature of materials as compared to conventional RGB cameras. Similar to adding color information to a grayscale image that requires recording the intensity in three different wavelength bands, a hyperspectral camera is able to capture information on the radiance of recorded materials in many, potentially non-visible, bands. This technology is widely used for satellite and airborne recordings, where the most common use cases are related to agriculture [1–3], vegetation monitoring [4–6] and mineralogy [7]. Another application for hyperspectral imaging is industrial food inspection [8,9]. The review of Adão et al. [1] lists several hyperspectral imaging sensors available on the market.

Bradley et al. [10] report a multispectral solution to detect vegetation for mobile robot navigation. They used an RGB camera and a monochrome camera with a near-infrared-pass filter to record four bands, however, frequently misclassified clothes as vegetation. Grönwall et al. [11] experimented with several active and passive camera sensors operating at different wavelengths to detect people in vegetation. Their work contains many visual comparisons between sensors and spectral signatures of

several types of clothes. Kandylakisa et al. [12] have built a platform with a thermal, short wave infrared (SWIR) and a hyperspectral camera for critical infrastructure monitoring. The developed multi-modal system is able to detect and track moving targets through fog and smoke. Puttonen et al. [13] used an active hyperspectral LiDAR and showed 92.3% accuracy for detecting camouflage versus vegetation.

Several authors have published applications of hyperspectral cameras, which we consider in our paper. Al-Sarayreh et al. [9] applied hyperspectral imaging to inspect red meat. Utilizing a deep convolution neural network, they classified between lamb, pork, beef and fat with an F1 score of 96.01%. Winkens et al. [14] applied hyperspectral imaging to terrain classification for autonomous driving where it is essential to know if the terrain is driveable. They compiled a database of hyperspectral samples, with three annotated labels. Their classifier for detecting obstacles achieved a precision of 70% and a recall of 56%. Cavigelli et al. [15] have applied a per-pixel deep fully connected network to classify spectral samples in an urban environment. They achieved an error rate of 9.4% with spectral measurements combined with RGB measurements for a multi-class (eight labels in total) classification. Furthermore, they applied a convolutional neural network to exploit spatial information with an error rate of 1.1%.

In this paper, we investigate the applicability of hyperspectral cameras—possibly mounted onto a drone—to detect people or vehicles in areas of dense vegetation. State-of-the-art object detection algorithms frequently fail in such scenarios because large parts of an object might be occluded by vegetation. With our approach, we aim to train a classifier that leverages a wide range of spectral information to identify fine-grained and possible incomplete areas of the image pertaining to people or vehicles. We use a lightweight hyperspectral camera that operates in the red and near-infrared (NIR) spectrum and returns intensities for 25 wavelength bands. We introduce a novel method to automatically annotate large amounts of hyperspectral samples based on state-of-the-art computer vision techniques. We train a deep neural network for classification and describe several image processing techniques to improve the accuracy of the classification. Finally, we detail quantitative results, show visual results, and draw a conclusion from them.

We pursue two main use cases: Search and rescue missions, and perimeter surveillance. In the former case, airborne fast through-foliage detection could be key to locate a person in distress or a fugitive (or their vehicle) by screening a large area with potentially challenging terrain. Likewise, through-foliage detection with hyperspectral cameras could be used to detect irregular migration and smuggling operations across the green border.

## 2. Materials and Methods

### 2.1. Radiometry

In this subsection we briefly reiterate a few radiometry terms based on Howell et al. [16], to facilitate a better understanding of the following sections. When radiant energy is being emitted, transmitted or received, it is called radiance—spectral radiance defines the energy per wavelength. The reflectance of a material describes the effectiveness of reflecting radiant energy. It is the ratio of the reflected and received radiance.

### 2.2. Existing Spectral Datasets

In satellite imaging, the Normalized Difference Vegetation Index (NDVI), as given by Equation (1), is often used to estimate the vegetation density on the earth's surface. The exact values for cutoff wavelengths of the near-infrared (NIR) and visible red (RED) bands in the spectrum depend on the use case and the sensor type. For example, Bradley et al. [10] chose to sum the reflectance from 600 to 670 nm for RED and 900 to 1000 nm for NIR. They used the NDVI, among other vegetation indices, to classify between vegetation and non-vegetation areas.

$$\text{NDVI} = \frac{(\text{NIR} - \text{RED})}{(\text{NIR} + \text{RED})} \tag{1}$$

Winkens et al. [14] published an annotated dataset collected with both XIMEA VIS (430 to 630 nm) and NIR (600 to 975 nm) snapshot cameras. The dataset does not contain a large number of samples and some material types have only been collected from a single object. Their NIR camera was equipped with a long-pass filter and operated in the range from 675 to 975 nm. Using their NIR dataset, we plot the NDVI features of selected samples in Figure 1. In our NDVI plot, we defined features RED and NIR as the sum of reflectance from the wavelengths 673 to 728 nm and from 884 to 944 nm, respectively.

Using the spectral reflectance database published by NASA [17,18], we can observe the reflectance of vegetation/soil vs. several man-made objects, e.g., asphalt, brick and metal. We have selected a few samples from the database and visualize them in Figure 2. Vegetation shows a higher reflectance at around 540 nm compared to the ranges from 400 to 500 nm and from 640 to 660 nm, where absorption by the chlorophyll takes place. It also shows a pronounced increase of reflectance around 700 nm. Grönwall et al. [11] performed a comprehensive comparison between active and passive sensors at different wavelengths. They also measured the spectral reflectance of several types of clothes. Their analysis shows that clothes exhibit a similar increase in reflectance around 700 nm.

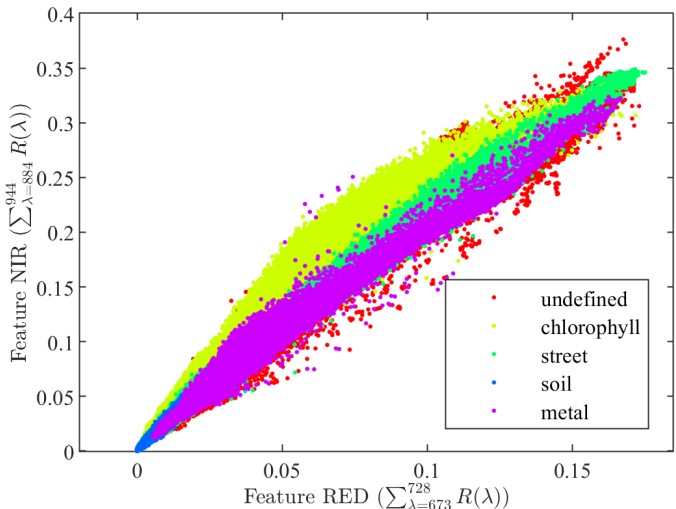

**Figure 1.** Normalized Difference Vegation Index (NDVI) features computed from the database of Winkens et al. [14].

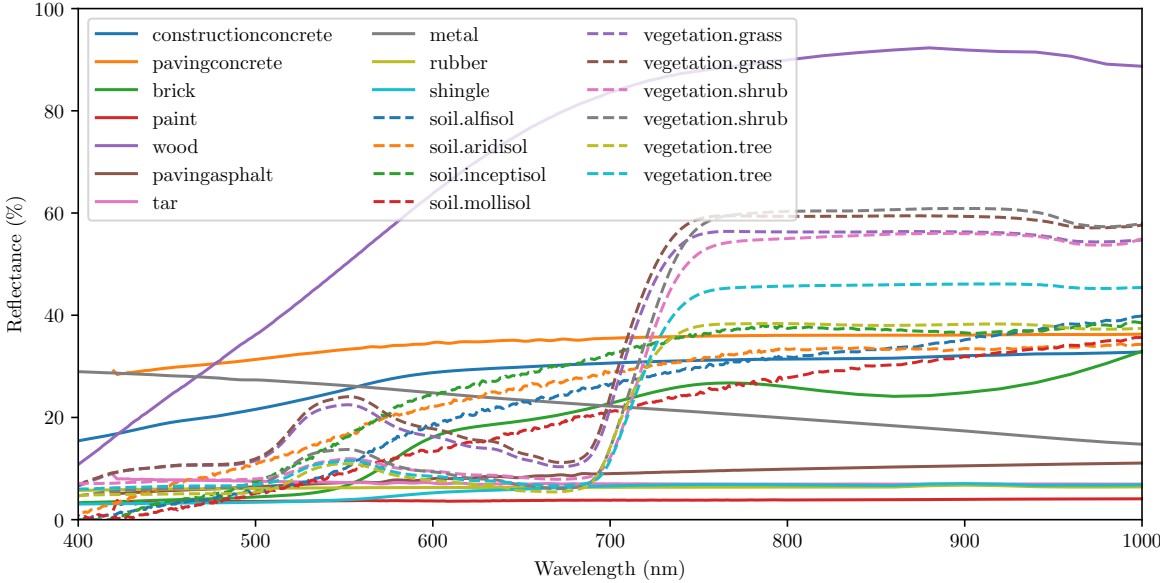

**Figure 2.** Reflectance of selected samples from the NASA database [17,18].

*2.3. Compact Passive Hyperspectral Cameras*

The current state-of-the-art solution for compact hyperspectral imaging has been developed by IMEC [19]. One of the companies that produce off-the-shelf cameras with this technology is XIMEA. The XIMEA MQ022HG-IM cameras weigh around 32 g and consume 1.6 W of energy. At the time we started developing our system, XIMEA provided a range of models as detailed in Table 1. For the MQ022HG-IM-SM5X5-NIR camera, an additional external optical filter was necessary to avoid double peaks, i.e., cross-talk between wavelength bands. From mid 2019—after we purchased our equipment—XIMEA started to introduce second-generation snapshot camera models and now offers solutions that do not require such an (external) filter. Additionally, XIMEA introduced a new model to specifically capture red and NIR portions of the spectrum from 600 to 860 nm, the MQ022HG-IM-SM4X4-REDNIR. In contrast to the other NIR hyperspectral cameras, it operates at a lower spectral (16 bands), but at a higher spatial (512 × 272 pixels/band) resolution.

**Table 1.** Selection of available XIMEA hyperspectral cameras for our system (Generation 1).

| Camera Type (MQ022HG-IM) | SM4X4-VIS | SM5X5-NIR | SM5X5-NIR |
|---|---|---|---|
| External Optical Filter | - | Short Pass | Long Pass |
| Wavelength (nm) | 465–630 | 600–875 | 675–975 |
| Pattern Size (pixels) | 4 × 4 | 5 × 5 | 5 × 5 |
| Spectral Resolution (bands) | 16 | 25 | 25 |
| Spatial Resolution (pixels/band) | 512 × 272 | 409 × 217 | 409 × 217 |

The concept of the IMEC technology is that on top of a CMOS image sensor, filters are deposited and patterned directly. Depending on the spectral bands, they are organized in 4 × 4 or 5 × 5 arrays in a mosaic pattern. Each array captures the spectral information of the incident light. After a raw frame has been captured, see Figure 3a, it needs to be demosaiced to receive a hyperspectral cube, see Figure 3b. Depending on the use case, the bands can be stored interleaved ($H(x, y, \lambda)$) or planar ($H'(\lambda, x, y)$), where $H$ and $H'$ is the hyperspectral cube.

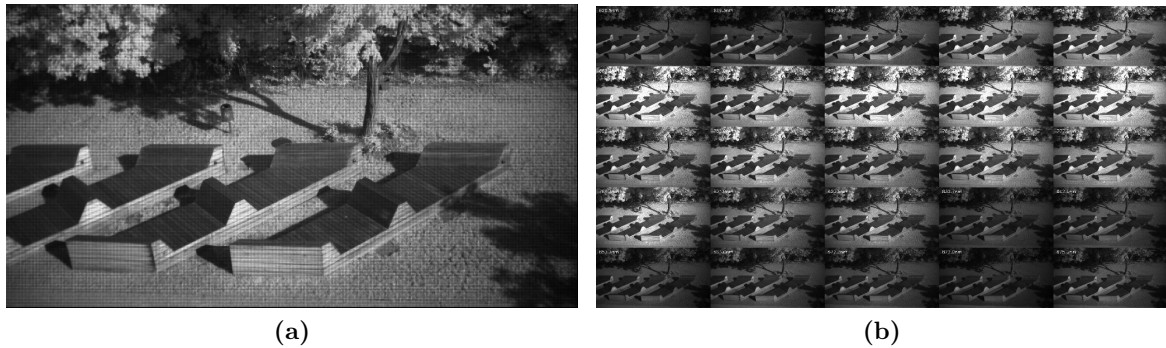

(a)                                          (b)

**Figure 3.** (**a**) Hyperspectral input frame and (**b**) planar hyperspectral cube after demosaicing.

2.3.1. Hyperspectral Frame to Hyperspectral Cube Conversion

The first step in data processing is to convert the raw hyperspectral frame (Figure 3a) to a hyperspectral cube (Figure 3b). Let us denote the raw hyperspectral input frame as a tensor $H_r$ with the indices $(x, \lambda_x, y, \lambda_y)$. Furthermore, let us denote the hyperspectral cube as a tensor $H$ with the indices $(x, y, \lambda)$, where

- $x$ is the horizontal spatial coordinate,
- $y$ is the vertical spatial coordinate,
- $\lambda_x$ is the horizontal raw spectral coordinate in one mosaic pattern array,
- $\lambda_y$ is the vertical raw spectral coordinate in one mosaic pattern array,

- $\lambda$ is the corrected spectral coordinate.

Let $H_c$ be the camera parameter correction tensor with the indices $(\lambda_x, \lambda_y, \lambda)$, which is obtained during the camera factory calibration. Performing a tensor contraction operation, we can define the equation $H := H_r \otimes H_c$, which computes the corrected spectral values and also reorganizes the values to a hyperspectral cube. In case we require a planar representation of the hyperspectral cube, we can define $H'$ with the indices $(\lambda, x, y)$ and perform the tensor contraction as $H' := H_c \otimes H_r$.

### 2.3.2. Overlapping Spectral Radiances

The filters on the CMOS sensor are organized in a $4 \times 4$ or $5 \times 5$ pattern, which captures the spectral information of the incident light. If more than one material contributes to the light received at a pattern, the spectral information represents, of course, a combination of those materials. Such inconsistency can appear when a mosaic pattern captures the border of two materials. This can lead to false classification due to the unreliable information.

To overcome this issue, we developed a filter method. It is based on the assumption that one material should exhibit a locally uniform spectral radiance, i.e., adjacent mosaic patterns covering the same material should deliver similar radiance values over all spectral bands. The core of this method resembles a standard edge detection, and positions, where a high difference value is detected, should not be considered in the classification process. We define this difference value $\Delta(x_0, y_0)$ at a position $(x_0, y_0)$ with respect to its neighborhood $V$ in Equation (2). Finally, we empirically select a threshold for the computed difference value. The size and shape of the neighborhood set $V$, and the threshold depends on the user's choice of filter strength.

$$\Delta(x_0, y_0) := \frac{1}{|V|} \sum_{(x,y) \in V} \sqrt{\sum_{(\lambda_x, \lambda_y) \in \Lambda} (H(x_0, \lambda_x, y_0, \lambda_y) - H(x, \lambda_x, y, \lambda_y))^2}. \tag{2}$$

### 2.3.3. Radiance to Reflectance Transformation

Spectral analysis is based on the reflectance of the material. Unfortunately, data acquisition with a passive hyperspectral camera is affected by several factors, see Table 2. The raw data from the camera are pixel intensities, which are affected by the camera characteristics. These parameters can be acquired in a laboratory after production for each camera, therefore eliminating this factor is straightforward. After the conversion from raw pixel intensities, the result will be radiance values.

**Table 2.** Factors affecting quantities obtained from passive hyperspectral camera systems.

| Raw Pixel Values | Radiance | Reflectance |
| --- | --- | --- |
| Light source | Light source | Material properties |
| Atmospheric absorption | Atmospheric absorption | |
| Camera characteristics | Material properties | |
| Material properties | | |

Radiance is mainly determined by a combination of material properties and the light source. The difficulty of eliminating the light source and obtaining reflectance depends on the set-up of the measurement. In a controlled environment (e.g., a conveyor belt in an industrial building), this can be done by measuring the minimum radiance (no light source) and the maximum radiance (light source with a calibration board) [9]. Calibration boards have a well-known reflectance in the spectral range in which the measurements will be executed. Once minimum and maximum measurable radiance are known, it is possible to calculate the reflectance of a material.

In aerial scenarios, the viewpoint limits the complexity of the scenario. This is because when looking towards the ground, illumination is generally evenly distributed and no direct sunlight will reach the sensor array. Two cases can be considered: Areas with direct sunlight and shadow.

The calibration can be done in a similar fashion as in a controlled environment. Prior to the flight, a calibration board is measured in direct sunlight [1]. Therefore, during the flight, the reflectance of objects in direct sunlight (e.g., tree canopy, soil) can be measured.

Terrestrial scenarios are more complicated, due to varying angles of incidence of the sunlight and generally more complex superposition of shadows. This makes it challenging to reliably convert radiance to reflectance in such scenarios. Winkens et al. [14] proposed to apply normalization on the radiance values and to detect shadows with thresholds. Alternatively, a normalization technique like max-RGB could help to obtain approximated reflectance values in some cases, e.g., when parts of the camera's field of view cover a highly reflective wall.

For demonstration, we have selected three materials from the scene in Figure 3. The materials are the wooden bench, grass and the metal waste bin. We performed a maximum radiance calibration with a calibration board (not seen in the image). Figure 4a shows the raw pixel intensities, which are received directly from the camera. After the elimination of the camera characteristics from the raw pixel intensities, we receive radiance in Figure 4b. Knowing the maximum radiance, we can approximate the reflectance of the objects in Figure 4c. It can be seen that the calibration was not perfect, as we get values higher than 1.0 for reflectance. Alternatively, we can calculate the normalized radiance in Figure 4d as proposed by Winkens et al. [14].

We conclude that, for a highly accurate classification, reflectance is an essential parameter. It solely depends on the material itself, not on ambient conditions. However, it is not always possible to estimate the reflectance of objects from hyperspectral images in uncontrolled environments.

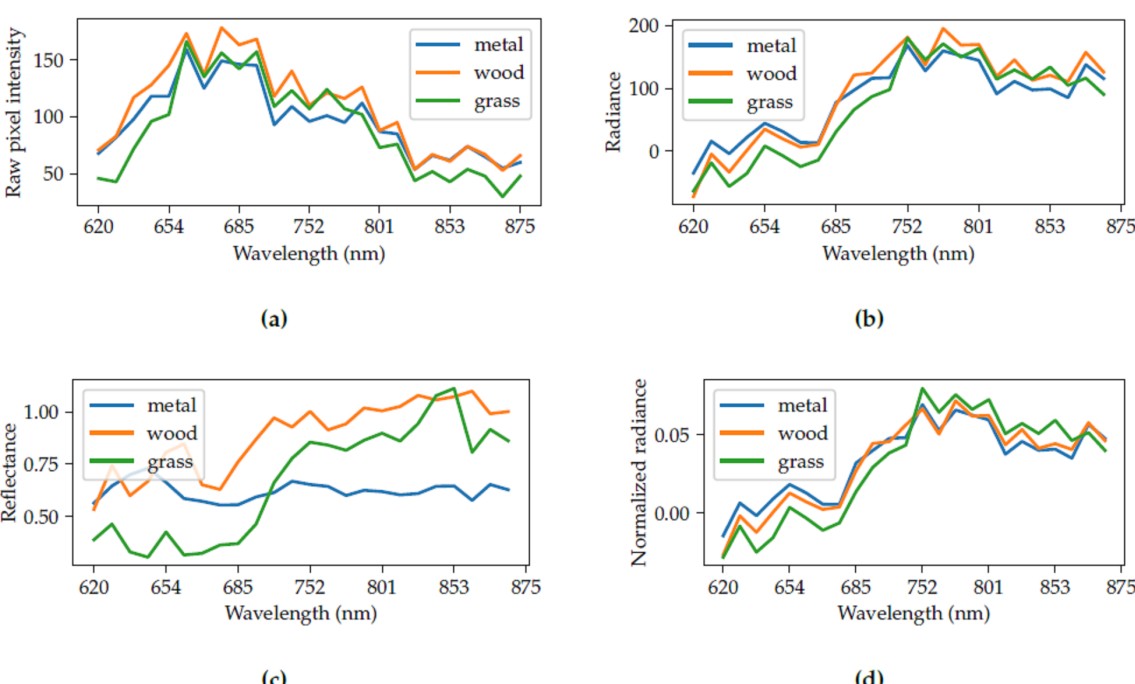

**Figure 4.** (**a**) Raw pixel intensity, (**b**) radiance, (**c**) reflectance, and (**d**) normalized radiance of different materials in a hyperspectral frame.

## *2.4. Data Acquisition*

### 2.4.1. Camera System

We selected the XIMEA MQ022HG-IM-SM5X5-NIR (Generation 1; XIMEA GmbH, Münster, Germany) hyperspectral camera with a short-pass filter because it offered the best trade-off in coverage of the visible and NIR bands (Table 1). The full width at half maximum of spectral bands ranges

between about 5 and 20 nm. Figure 5a shows the compact casing of the MQ022HG-IM series compared to a 10 Euro Cent coin.

The hyperspectral camera was mounted in a custom-made multi-purpose stereo camera housing, see Figure 5b. The housing also contained an FLIR Boson thermal camera and a FLIR Blackfly 4K RGB camera, however, we did not use the thermal camera in this study. We developed a custom recording software to store the frames from all cameras semi-synchronously. No hardware trigger was available with the current setup. Therefore, each camera was recorded in a separate thread and the timestamp of the retrieval of the frame was saved for synchronization.

Figure 5c visualizes the response curves for each band of the XIMEA MQ022HG-IM-SM5X5-NIR. The filter transfer function indicates the selected wavelength range.

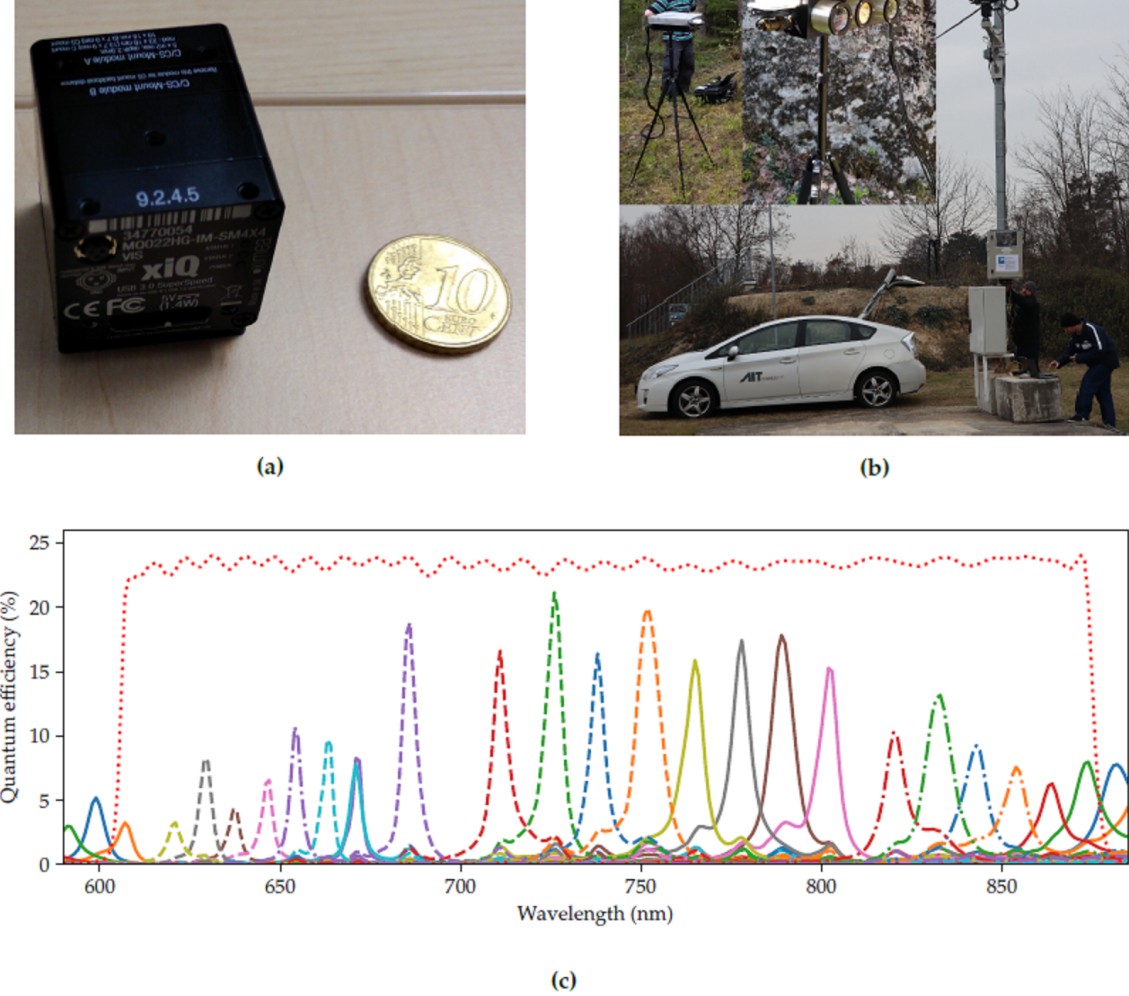

**Figure 5.** The camera system. (**a**) XIMEA MQ022HG-IM hyperspectral snapshot camera series compared to a 10 Euro Cent coin. (**b**) Camera system on a tripod (inlays, top-left), and on a mast equipped with a pan-tilt unit. (**c**) Response curves of the XIMEA MQ022HG-IM-SM5X5-NIR (Generation 1). The red dotted line is the combined transfer function of the optical filters.

### 2.4.2. Recording Scenarios

We distinguish between three major types of recordings (four recordings in total), which are summarized in Table 3. For all recordings, the camera field of view remained static. The first recording (Ground) was performed on the ground, and can be considered as a terrestrial scenario. The second recording (Quarry) was performed at an abandoned quarry, where the camera was mounted at

a height of around 50 m. This recording is a simulation of an airborne scenario. For recordings 1 and 2, the hyperspectral camera was set to the constant exposure time. We recorded short scenes with and without actors being present. The actors moved around either by walking or crouching, or stood still. They were either in the open or hid behind trees, bushes or lay on the ground surrounded by high grass. The scenes were not scripted because we did not want to detect specific actions but simply the presence of people. Recordings 3 (Mast October) and 4 (Mast November) are terrestrial scenarios too. The camera system was installed on a 4-meter high mast for long-term outdoor recording as shown in Figure 5b. We opted for this installation because it provided the necessary infrastructure to facilitate long-term unsupervised recordings such as power supply and a waterproof box to house the recording computer, and it provided some protection against vandalism and accidental damages. We used a YOLO detector [20] to continuously monitor the video stream and only record scenes where a person, car or truck were present. To cope with changing illumination conditions and the difficulties mentioned in Section 2.3.3, we used varying exposure times for the hyperspectral camera. We explain this concept in more detail in Section 2.4.4.

**Table 3.** Overview of the recording scenarios.

| Name | Runtime Days | Recorded Minutes | Comments |
| --- | --- | --- | --- |
| Ground | - | 10 | Manual annotation, single exposure |
| Quarry | - | 15 | Manual annotation, single exposure |
| Mast October | 21 | 350 | Auto annotation, multiple exposure |
| Mast November | 18 | 142 | Auto annotation, multiple exposure |

### 2.4.3. Vegetation

Throughout this paper, the term vegetation refers to all plants or plant material captured by the camera. This includes leaves, branches, stems etc. of living plants, but also e.g., dead branches lying on the ground. The types of plants range from grass to bushes and trees.

The recordings were carried out in Central and Eastern Europe, featuring vegetation that is typically found at this latitude. In the case of the Ground dataset, the vegetation comprised of grass and other lowbush plants, and conifers, mostly spruce trees. The Quarry dataset, on the other hand, mostly features deciduous trees and bushes alongside approximately knee-high grass. Finally, the Mast datasets contain grass, a deciduous tree and bushes.

### 2.4.4. High Dynamic Range for Hyperspectral Images

As described in Section 2.3.3, one frame can show both overexposure (i.e., saturation) and underexposure (i.e., low Signal to Noise Ratio (SNR)) at the same time. One approach to deal with this challenge is to collect a series of frames with varying exposure times and fuse them into one well-illuminated frame. This improves the local exposure of the target frame (Figure 6) at the expense of some additional motion blur. Figure 6a–c show three frames of the same scene aquired with different exposure times ranging from 1 to 4 ms.

We use the following method to select appropriate mosaic patters from the respective frames: Let us denote the set of exposure times as $E$ and for each $e \in E$, the corresponding raw hyperspectral input frame as $H_e$. The set of raw spectral coordinates is defined as $\Lambda$. First we compute the maximum spectral value $m_e(x, y)$ for each frame $H_e$, as defined in Equation (3). Then, for each pair of coordinates $(x, y)$, we choose the index of exposure $p(x, y) \in E$ which fulfills Equation (4). We define a spectral measurement (mosaic pattern) as valid if it does not contain any saturated value (i.e., a raw pixel value of 255). Finally, we build up one high dynamic hyperspectral raw frame, denoted as a tensor $H_{\mathrm{HDR}}$, as defined by Equation (5). Figure 6d shows how the final frame is constructed.

$$m_e(x, y) := \max\{H_e(x, \lambda_x, y, \lambda_y) \mid (\lambda_x, \lambda_y) \in \Lambda\}, \, e \in E \tag{3}$$

$$m_{p(x,y)}(x,y) = \max\{m_e(x,y) \mid e \in E \wedge m_e(x,y) < 255\} \tag{4}$$

$$H_{\text{HDR}}(x, \lambda_x, y, \lambda_y) := H_{p(x,y)}(x, \lambda_x, y, \lambda_y) \ , \ (\lambda_x, \lambda_y) \in \Lambda \tag{5}$$

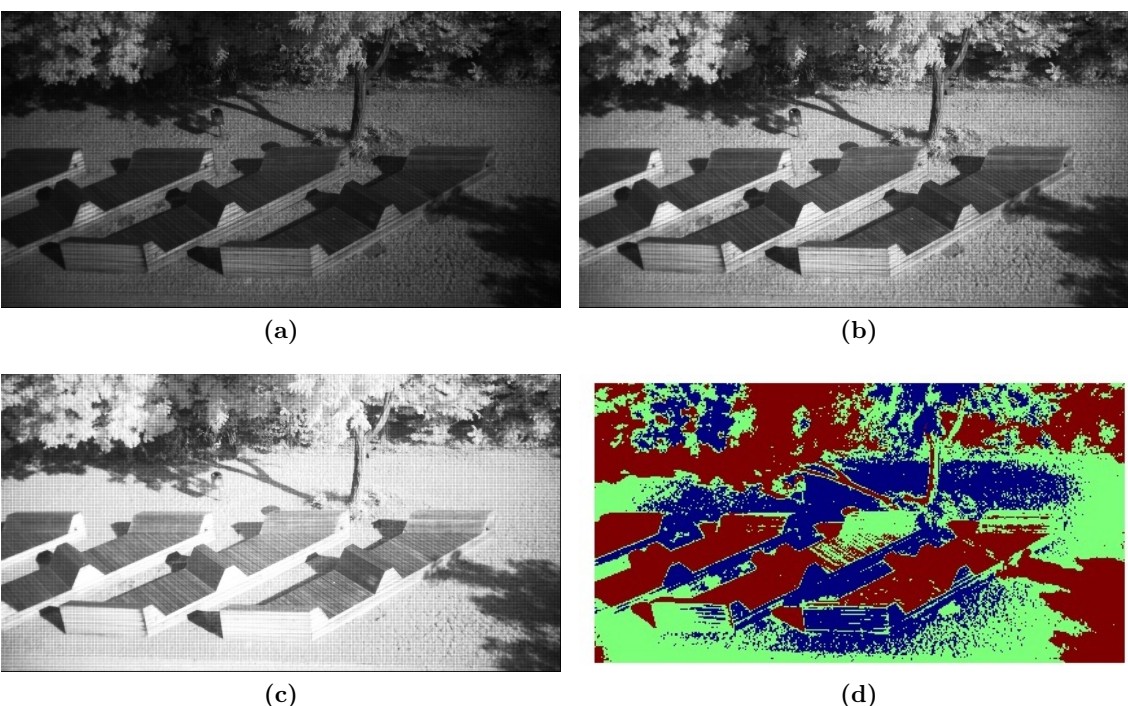

**Figure 6.** Visualization of our high dynamic range (HDR) solution for hyperspectral frames with exposure times of (**a**) 1 ms, (**b**) 2 ms, and (**c**) 4 ms. (**d**) Combined HDR mask with mosaic patterns selected from (**a**) blue; (**b**) green; and (**c**) red.

### 2.5. Automatic Annotation of Hyperspectral Frames

We applied an automatic method to annotate the hyperspectral frames. The pseudocode of the algorithm is given in Algorithm 1. Each hyperspectral band can be visualized as one grayscale image, resulting in 25 grayscale images. On each grayscale image, we run a MaskRCNN algorithm [21], see Figure 7a. The MaskRCNN, in addition to bounding boxes, delivers a pixelwise classification with confidence values. Finally, the 25 pixelwise classifications are merged together based on thresholds, which results in one compound mask per hyperspectral frame per object, see Figure 7b. The thresholds, which are applied to the confidence values of the MaskRCNN results, can vary based on the user's choice of algorithm sensitivity. In the merging process, thresholds are applied to the mean of the remaining confidence values per band. The resulting mask allows us to automatically retrieve and label individual spectral radiance samples. We qualitatively assessed the robustness and reliability of this approach by manual review of typical scenes. The visual impressions were satisfactory and hence we accepted this approach going forward.

### 2.6. Spectral Signal Classification

We trained a deep neural network as a binary classifier to decide whether a spectral radiance sample represents background (e.g., vegetation or tarmac) or a foreground object (e.g., textile, plastic or metal). As proposed by Cavigelli et al. [15], the deep neural network has six fully connected layers with five consecutive ReLu activation functions and one SoftMax activation for the last layer, see Figure 8. Each sample was presented to the network in vector form. In contrast to Cavigelli et al., we could not rely on structural information, because the objects can be partly occluded by vegetation.

Therefore, we trained the classifier on spectral radiance measurements only. The feature values were the corrected radiance values, as described in Section 2.3.3. Additionally, we repeated the analysis with the normalized radiance and added the normalization factor to the feature vector. The normalization was performed as proposed by Winkens et al. [14].

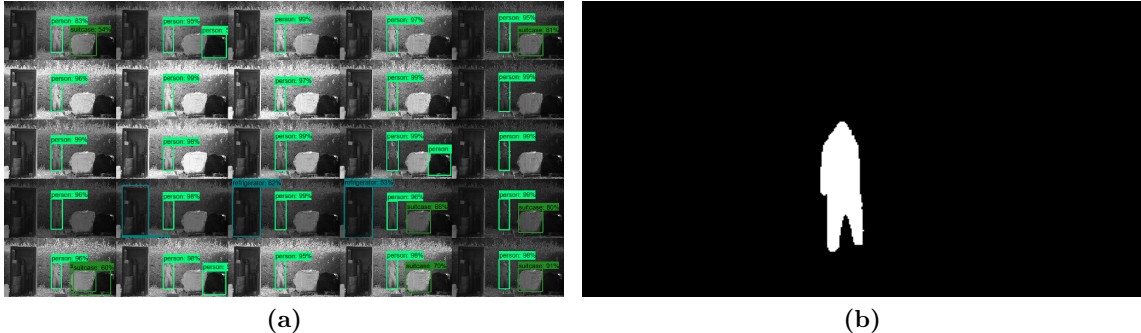

<center>(a)            (b)</center>

**Figure 7.** Automatic annotation: (**a**) MaskRCNN object detection in 25 spectral bands. (**b**) Merged mask of all MaskRCNN detections (magnified).

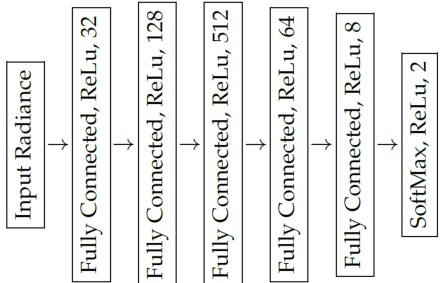

**Figure 8.** Layout of the deep neural network applied for classification.

---

**Algorithm 1:** Automatic annotation of hyperspectral frames.

---

**Result:** matrix of spectral radiance samples
load raw hyperspectral frame;
demosaic frame to planar raw hyperspectral cube;
**for** *each band in raw hyperspectral cube* **do**
     perform detection with MaskRCNN;
     threshold pixelwise detection based on the confidence values to get a mask for the band;
**end**
merge band masks based on thresholds;
retrieve spectral radiance from raw hyperspectral cube based on merged mask;

---

We defined a sample as the spectral radiance at one $(x, y)$-position in a hyperspectral cube $H$, as defined in Section 2.3.1. We called samples pertaining to person/car/truck/clothes objects positive and tarmac/tree/vegetation objects negative samples.

We used our automatic annotation method to collect positive samples (person, car, truck) in the Mast October/November datasets. The automatic annotation was able to collect samples only where no occlusion was present. This enabled us to train and test a classifier, and later perform visual evaluation where occlusion appeared.

In the Ground dataset, we manually annotated clothes as positive samples. To track individual pieces of clothes, we added a consecutive number to the respective labels, e.g., clothes-clothesid-01.

The number was assigned in the order in which the clothes appeared in the video and does not indicate any specific characteristics of the fabric.

During the first manual annotations, we quickly noticed that occluded samples were nearly impossible to annotate manually. Therefore, manual annotation was only applied to non-occluded positive samples.

Negative samples (e.g., vegetation) were annotated manually in all datasets, except in the Mast November dataset. The Mast October and Mast November datasets both show the same location over an extended period of time with varying illumination and weather conditions. We concluded, that Mast November does not introduce much additional variation and would not change the statistics significantly. Hence we did not collect negative samples from it. There was only one tree in the field of view of the Mast October dataset, which is the reason for the low number of tree samples.

In the Ground and Quarry datasets, we introduced a simple taxonomy to study the performance of our classifier on different types of vegetation. First, we discriminated between forest (mix of standing trees, large bushes) and ground (generally low vegetation and little shadow, e.g., in clearances or open fields). We then evaluated the leaf density of forest areas ranging from low (very sparse leaves) to high (very dense leaves, not present in the data). In case of ground areas, we defined patches with a vegetation height up to 10 cm as low, and up to 30 cm as medium.

Training was performed on the Mast October dataset, where we obtained 6,198,185 positive and 2,475,260 negative samples. To maintain a balanced dataset we randomly selected 2,475,260 samples from the positive set. Finally, the data was split into 3,960,416 (80%) training samples and 990,104 (20%) validation samples. The test dataset was Mast Novemeber, which contains 11,439,949 positive samples. We did not explicitly train on clothes samples or persons seen in the Quarry or Ground datasets. In these instances, we accepted a positive classification, i.e., not vegetation/ground, regardless of the specific class of each sample.

To classify unseen data, each radiance sample was evaluated by the neural network. We thus obtained a confidence value (from 0 to 100%) indicating whether the evaluated sample belongs to a person/vehicle. The classification result with the higher confidence value was accepted.

### 2.7. Implementation and Workflow

The algorithm was implemented in Python with NumPy and TensorFlow [22]. Demosaicing and raw pixel values to radiance computation can be performed efficiently in one step with the tensordot function. The inference is done with TensorFlow, which allows for both CPU and GPU execution. Since our neural network is lightweight, we did not see a substantial difference in execution speed between CPU and GPU. Demosaicing and inference can be executed with around 10 frames per second on a conventional workstation (Intel(R) Xeon(R) W-2145 CPU @ 3.70 GHz). High dynamic range and suppression of overlapping spectral radiances have not been optimized, only Python prototypes were tested. Figure 9 summarizes the main steps of our workflows.

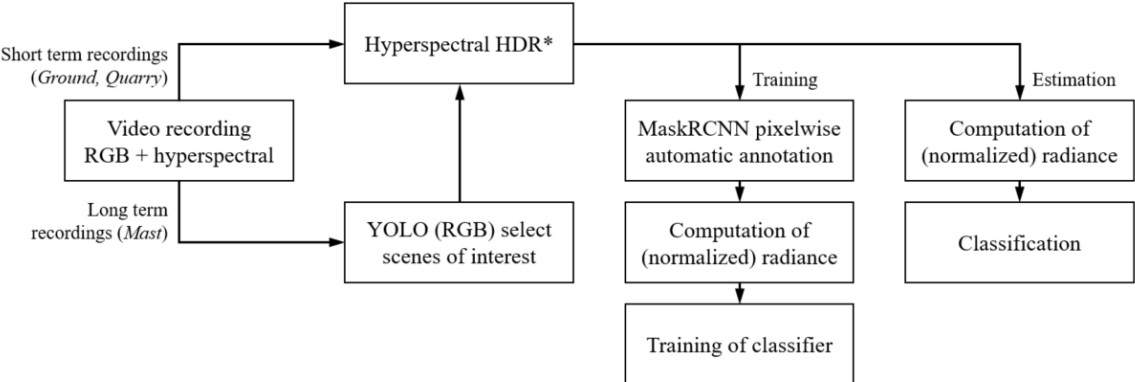

**Figure 9.** Main steps of the training and estimation workflows. * Hyperspectral HDR was only available for the Mast datasets. HDR = High Dynamic Range.

## 3. Results

As mentioned in Section 2.6, our objective was to detect a person, car or truck behind vegetation or under foliage. For this purpose, we applied a classification based only on the spectral distribution and no structural information was used. In this section, we present a detailed quantitative analysis, followed by visual inspection based on two examples.

### 3.1. Quantitative Results

Table 4 shows the classification results applied to the positive samples collected by automatic annotation. The samples from Mast November were not used during training. Both radiance and normalized radiance results are compared. The results show that person samples were correctly classified with a sensitivity of up to 95.5% and car samples with a sensitivity of up to 86.6%. While person samples were classified about equally well in both the Mast October and Mast November datasets, we observed a considerable drop of about 10 to 15% sensitivity in case of car and truck samples in the test dataset (Mast November). Classification based on normalized radiance yielded generally better results compared to radiance. The margin of improvement ranged from about 2 to about 7% sensitivity. Interestingly, the Mast November car and truck samples benefitted the most from normalizing spectral radiance.

In Table 5, classification results of manually annotated samples are shown. For the Mast October dataset, we were able to classify negative samples with high specificity. On a count-weighted basis, the overall specificity reached 97.3% and 96.2% in case of radiance and normalized radiance, respectively. In contrast to the classification of positive samples from the same dataset, normalized radiance lead to marginally inferior performance.

The Quarry forest-density-low samples, where it is possible to see through sparse leaves but illumination is strongly diminished, were classified with a relatively low specificity of 83.8%. The specificity was considerably lower if normalized radiance was used (33.0%). For the samples forest-density-medium, where the trees have more leaves and fewer openings are present, most of the samples are classified correctly as negative samples. The classification results of the sample ground-vegetation-medium show that properly illuminated ground covered in vegetation can be classified correctly.

When observing the negative samples from the Ground dataset, one can see a markedly lower classification specificity compared to the Quarry dataset. The best result was obtained for ground-vegetation-low samples classified by radiance (91.7% specificity). In case of both ground-vegetation-medium and forest-density-medium samples, the performance was obviously inferior. Classification based on normalized radiance produced even poorer results, especially in case of forest-density-medium, where we were only able to obtain a specifity of 53.6%.

Classification of clothes samples delivered mixed results. For example, clothes-clothesid-20 was classified with 99.4% sensitivity, whereas clothes-clothesid-16 only reached 4.4% sensitivity. Radiance normalization generally dimished classification sensitivity, except for clothes-clothesid-20.

Finally, we evaluated the classifier using the Matthews Correlation Coefficient (MCC) which we plot in Figure 10. For a naive classification threshold of 50%, we were able to obtain MCC of 0.78 and 0.83 in case of radiance and normalized radiance, respectively. At optimized thresholds, the MCC would have been 0.83 and 0.85, respectively. The test dataset was composed of the positive samples from the Mast November and negative samples from the Mast October dataset.

**Table 4.** Classification results of automatically annotated data. TP = True Positive, FN = False Negative.

| | | | | | |
|---|---|---|---|---|---|
| **Mast October Dataset—Positive Samples** | | | | | |
| | **Radiance** | | **Norm. Radiance** | | |
| **Label** | **TP(%)** | **FN(%)** | **TP(%)** | **FN(%)** | **Count** |
| car | 93.44 | 6.56 | 96.40 | 3.60 | 1,433,115 |
| person | 94.43 | 5.57 | 96.42 | 3.58 | 3,209,489 |
| truck | 91.72 | 8.28 | 94.63 | 5.37 | 1,555,581 |
| **Mast November Dataset—Positive Samples** | | | | | |
| | **Radiance** | | **Norm. Radiance** | | |
| **Label** | **TP(%)** | **FN(%)** | **TP(%)** | **FN(%)** | **Count** |
| car | 79.74 | 20.26 | 86.64 | 13.36 | 8,407,865 |
| person | 93.12 | 6.88 | 95.49 | 4.51 | 641,724 |
| truck | 75.48 | 24.52 | 82.41 | 17.59 | 2,390,360 |

**Table 5.** Classification results of manually annotated data. FP = False Positive, TN = True Negative, TP = True Positive, FN = False Negative.

| | | | | | |
|---|---|---|---|---|---|
| **Mast October Dataset—Negative Samples** | | | | | |
| | **Radiance** | | **Norm. Radiance** | | |
| **Label** | **FP(%)** | **TN(%)** | **FP(%)** | **TN(%)** | **Count** |
| tarmac | 1.31 | 98.69 | 3.05 | 96.95 | 1,258,540 |
| tree | 3.89 | 96.11 | 3.94 | 96.06 | 11,872 |
| vegetation | 4.11 | 95.89 | 4.66 | 95.34 | 1,204,848 |
| **Quarry Dataset—Negative Samples** | | | | | |
| | **Radiance** | | **Norm. Radiance** | | |
| **Label** | **FP(%)** | **TN(%)** | **FP(%)** | **TN(%)** | **Count** |
| forest-density-low | 16.17 | 83.83 | 66.98 | 33.02 | 598,825 |
| forest-density-medium | 5.60 | 94.40 | 2.06 | 97.94 | 642,958 |
| ground-vegetation-medium | 0.05 | 99.95 | 0.17 | 99.83 | 2,066,311 |
| **Ground Dataset—Negative Samples** | | | | | |
| | **Radiance** | | **Norm. Radiance** | | |
| **Label** | **FP(%)** | **TN(%)** | **FP(%)** | **TN(%)** | **Count** |
| forest-density-medium | 35.17 | 64.83 | 46.40 | 53.60 | 141,431 |
| ground-vegation-low | 8.35 | 91.65 | 20.09 | 79.91 | 97,074 |
| ground-vegetation-medium | 20.36 | 79.64 | 24.98 | 75.02 | 959,979 |
| **Ground Dataset—Positive Samples** | | | | | |
| | **Radiance** | | **Norm. Radiance** | | |
| **Label** | **TP(%)** | **FN(%)** | **TP(%)** | **FN(%)** | **Count** |
| clothes-clothesid-11 | 56.77 | 43.23 | 47.95 | 52.05 | 11,148 |
| clothes-clothesid-16 | 4.44 | 95.56 | 3.24 | 96.76 | 4258 |
| clothes-clothesid-18 | 74.95 | 25.05 | 70.16 | 29.84 | 9054 |
| clothes-clothesid-20 | 99.39 | 0.61 | 99.93 | 0.07 | 12,142 |
| clothes-clothesid-24 | 29.86 | 70.14 | 26.44 | 73.56 | 7542 |

*3.2. Visual Results*

In Figure 11, a frame from the Quarry dataset simulating an aerial recording is shown. Figure 11b shows the RGB image of the corresponding input hyperspectral frame in Figure 11a. The classification results can be seen in Figure 11c,d without and with radiance overlap suppression, respectively. In this example, the forest-density-medium samples pertain mostly to the tree canopies and dense bushes on the ground. The forest-density-low samples cover less dense areas, for example, lower parts of the trees where generally fewer leaves can be found. The ground-vegetation-medium samples mostly represent the open field with high grass in direct sunlight. Ideally, only the two crouching persons in the center of the frame should be classified as positive samples, however, many samples have been misclassified. As also indicated by Table 5, forest-density-low samples showed the lowest specificity, predominantly in areas with low illumination. Samples of forest-density-medium, on the other hand, did relatively well, especially after spectral overlap supression was applied. Only a few ground-vegetation-medium samples were misclassified and the two persons are clearly visible in the classified frame. Evidently, the spectral overlap supression reduces misclassification in vegetated areas, however, less positive samples are also detected (see samples pertaining to persons in Figure 11c,d. This is because the filter partially erodes patches of the same material and thus decreases their overall size.

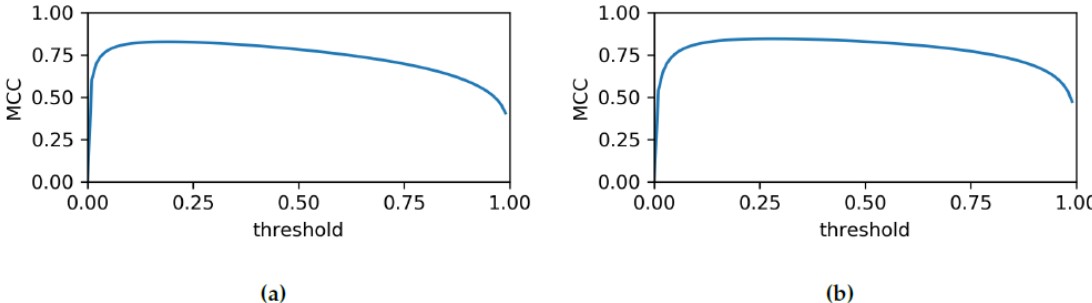

(a)                                         (b)

**Figure 10.** Matthews Correlation Coefficient (MCC) for varying classification thresholds based on (**a**) radiance, and (**b**) normalized radiance.

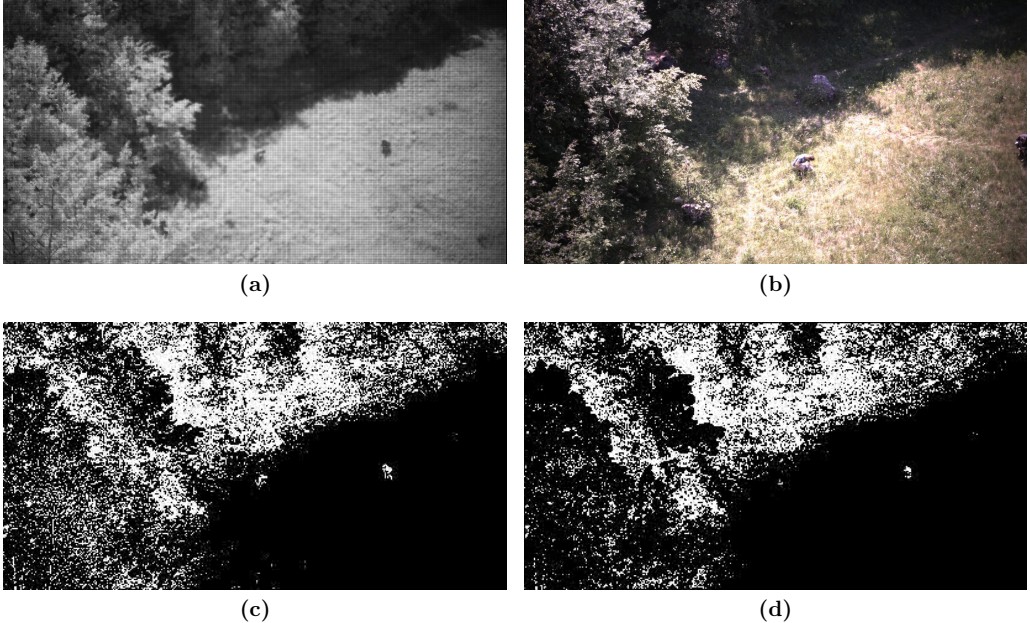

(a)                                         (b)

(c)                                         (d)

**Figure 11.** Aerial scenario (Quarry dataset), classification based on radiance: (**a**) Raw hyperspectral input frame. (**b**) Corresponding RGB frame. (**c,d**) Classification without and with spectral overlap supression, respectively. White: Classified as positive sample. Black: Classified as negative sample.

Figure 12 shows a frame from the Ground dataset, a terrestrial recording. Figure 12b shows the RGB image of the corresponding input hyperspectral frame in Figure 12a. The classification results can be seen in Figure 12c,d, without radiance overlap suppression and with suppression, respectively. The results in this scenario are less satisfactory as more false classifications are present.

Contrary to the aerial scenario, the ground samples yielded more false positives. This is consistent with the labels ground-vegetation-medium and ground-vegetation-low in Table 5. The same is true for samples from forest-density-medium. Not all clothes were classified as positive samples. For example, the clothes of the person closest to the camera are detected fairly well (albeit with some errors), whereas in the case of the second closest person, most parts of the shirt are not detected. The two persons relatively far away from the camera are almost indistinguishable from their surroundings in Figure 12c,d. The jacket of the person next to the camera is a good example to demonstrate the spectral overlap suppression filter. It not only discriminates the jacket against the vegetation in the background, but also against the pants, as indicated by the gap between the two clothes. It does not, however, break up the integrity of the jacket itself. Unfortunately, the sky is predominantly misclassified as a positive sample.

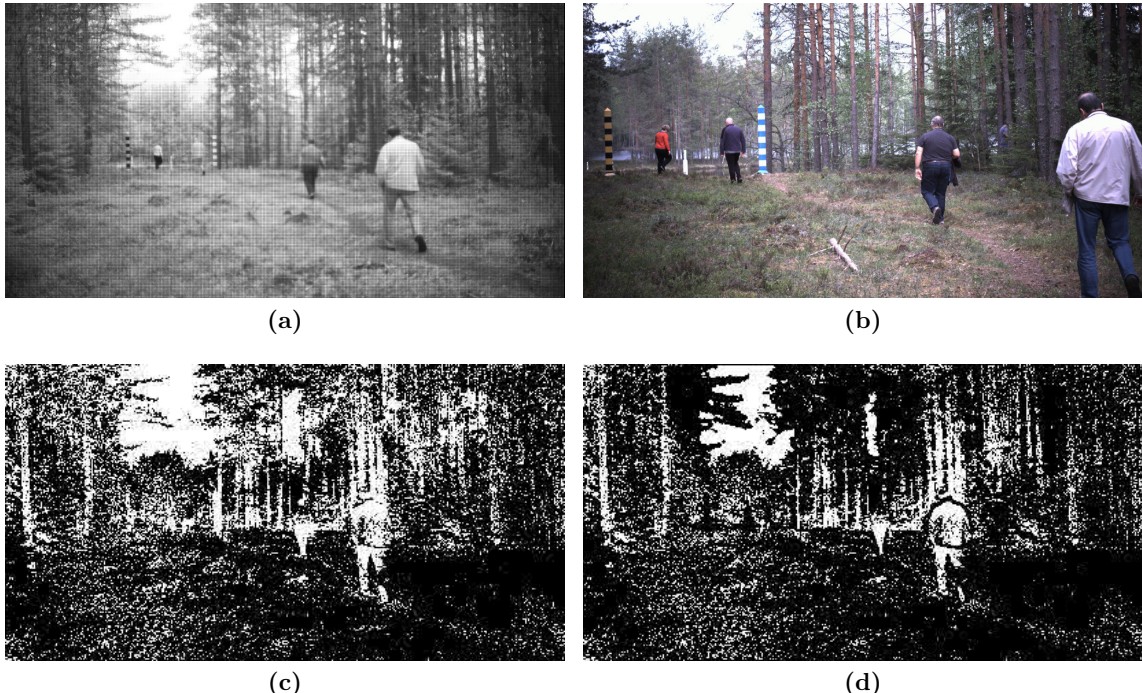

**Figure 12.** Terrestial scenario (Ground dataset), classification based on radiance: (**a**) Raw hyperspectral input frame. (**b**) Corresponding RGB frame. (**c**,**d**) Classification without and with spectral overlap supression, respectively. White: Classified as positive sample. Black: Classified as negative sample.

## 4. Discussion

The developed classifier shows generally good performance with an MCC of up to 0.83 (0.85 for optimized classification threshold) when evaluated on the Mast October and Mast November datasets. The latter dataset was not used for training, hence we used it as a test dataset. Using optimized thresholds would have only boosted the classification performance by 2% at the expense of introducing another tunable parameter. Table 4 shows, that the classifier generalized well for person samples, where specificity only dropped by about 1%. In case of car and truck samples, the classifier performed worse on unseen data. At this point, we can only speculate about the root cause. One explanation might be that (in the Northern hemisphere) in November the positon of the sun is generally lower than in October. Together with the reflective properties of car paint and windows, this could have

misled the classifier. The lower number of person samples in the Mast November dataset is most likely explained by seasonal variations as well: From visual inspection, we know that many detected persons were runners and in colder months less people tend to go running.

Our automatic annotation method proofed very useful to collect a large number of training samples. Nevertheless, it remains challenging to detect objects or persons through foilage. Using advanced object detection algorithms such as MaskRCNN [21] it is possible to obtain pixelwise spectral information (samples) and this can be used to classify samples pertaining to objects partially hidden by vegetation. We showed that it is possible to discriminate between persons and vehicles, and typical background in non-urban locations. However, we did not focus on the further classification of positive or negative samples. This leaves some open questions for future research. Moreover, the MaskRCNN algorithm was initially developed on RGB images and hence might show inferior performance on hyperspectral images. We mitigated this potential threat by establishing consensus across all 25 bands and obtained satisfactory results after qualitative visual inspection. Detailed quantitative analysis and potential data-driven parameter optimization remains a field for future research.

In the case of the Quarry and Ground datasets, the classification results were clearly inferior to the Mast datasets. Table 5 shows varying degress of specificity for different sample groups. The Quarry dataset was recorded from a relatively high vantage point—simulating an aerial scenario—in bright sunlight. A comaprison of the quantitative results and visual material in Figure 11 indicates, that most misclassifications of vegetation occur in areas with low illumination and that the type of vegetation might only be a secondary factor. This is further supported by Figure 12 (Ground dataset), where no direct sunlight illuminates the scene. Irrespective of the assigned label, the classifier performed worse than in case of the Quarry dataset. This lack of illumination could also explain the gap between the classification performance on the Mast and Quarry/Ground datasets. In case of the Mast datasets, we applied our hyperspectral HDR method, unfortunately, it was not available for the other recordings. While probable, this assumption should be tested in further research. We also showed in Figure 5 that the used XIMEA MQ022HG-IM-SM5X5-NIR (Generation 1) had a maximum quantum efficiency of between 20% and 25%. An improvement in quantum efficiency would also allow a better SNR at shorter exposure times.

As a consequence of the hyperspectral camera's relatively low spatial resolution of $409 \times 217$ pixels, the detection of small (parts of) positive samples is a challenging undertaking. With increasing distance, the spectral information of an object or person is projected to fewer pixels/samples, which decreases the classification accuracy. As described in Section 2.3.2, mosaic patterns covering more than one material are potentially unreliable due to spectral overlaps, i.e., the obtained spectrum will be a combination of two or more material's spectra. We hence developed an enhanced edge detection filter to suppress such samples. The snapshots in Figures 11 and 12 show that the filter is effective in suppressing ambiguous samples and can clearly differentiate between materials with different spectral signatures (Figure 12d). However, the filter also erodes patches of homogenous materials and consequently imposes further constraints on the minimum size of visible parts of the entities that should be detected. It would be possible to equip the hyperspectral camera with an objective that has a longer focal length to magnify observed objects, but this would decrease the field of view. With a future increase in sensor resolution, we see the potential for an increase in classification accuracy.

Varying classification sensitivity of clothes samples is most likely caused by different spectral signatures of different materials, as shown by Grönwall et al. [11]. For example, in case of clothes-clothesid-16, the spectral signature seems to be almost identical to the signature of vegetation. It is hence possible that our classifier could miss a person in some cases, even if the person would be very close to the camera.

In some cases, training and classification based on radiance yielded better results, and in some cases, normalized radiance was the better choice. When focusing on the Mast datasets, the normalized radiance seemed to render the classification more sensitive, but less specific. Overall, the MCC were of the same magnitude (radiance: 0.78, normalized radiance: 0.82), but normalization helped to improve

the classification; see Figure 10). Using normalized radiance seemed to diminish the classifier's performance in case of the Quarry and Ground in areas of suboptimal illumination conditions. Again, this points towards the relevance of our hyperspectral HDR method to obtain well-illuminated frames for reliable classification.

## 5. Conclusions

In this paper, we presented a novel automatic annotation technique for hyperspectral images in uncontrolled environments utilizing state-of-the-art deep learning methods. We trained a deep neural network that takes a pixelwise spectral sample as input and can classify between vegetation and people, vehicles, or clothes. The developed classifier was able to run in real-time on a conventional workstation. Its lightweight design serves our use cases—search and rescue missions, and perimeter surveillance in dense vegetation—where relatively inexpensive and readily available hardware is desirable. Additionally, a resource-light implementation lowers entry barriers for processing on mobile devices, so that it can be used directly in the field.

We evaluated our classifier on our dataset and obtained an MCC score of 0.83. However, correct classification strongly depends on proper illumination. Therefore, we developed a high dynamic range solution for hyperspectral recordings. Furthermore, one of the most prominent challenges remains the accurate transformation from radiance to reflectance in uncontrolled environments. To overcome this issue, one would have to either use active illumination or develop advanced methods to estimate the strength of the light source. There is ongoing research on deep learning techniques to approximate local contrast normalization factors for RGB images. In our future research, we will take this into consideration and would like to explore transferability to hyperspectral imaging. The currently available compact hyperspectral cameras operate at a relatively low spatial resolution (e.g., $409 \times 217$ pixels in case of the used XIMEA MQ022HG-IM-SM5X5-NIR (Generation 1)). With a future increase in sensor resolution, we see the potential for an increase in classification accuracy as well as working distance or field of view.

**Author Contributions:** Conceptualization, A.P.; data curation, A.P.; methodology, A.P. and J.P.; software, A.P. and J.P.; resources, D.B.; visualization, A.P.; writing—original draft preparation, A.P.; writing—review and editing, P.T. and C.W.; project administration, A.K.-Z. All authors have read and agreed to the published version of the manuscript.

**Funding:** This research was jointly carried out in the scope of two projects. FOLDOUT: This project has received funding from the European Union's Horizon 2020 research and innovation programme under grant agreement No 787021 ▪. The project AREAS is funded by the Austrian security research programme KIRAS of the Federal Ministry of Agriculture, Regions and Tourism.

**Conflicts of Interest:** The authors declare no conflict of interest.

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
