# Peer review of "Automatic Annotation of Hyperspectral Images and Spectral Signal Classification of People and Vehicles in Areas of Dense Vegetation with Deep Learning"

_remotesensing, doi:10.3390/rs12132111_

Round 1

Reviewer 1 Report

This manuscript is one of application of hyperspectral image processing foe the vegetation. 

However, the article is essential to improve the explanation of methodology and result. 

1) Section 2.1 : The definition of radiance and other physical properties are not essential. This section has to be deleted. Instead of this section, some definition of vegetation is essential. 

2) Table 2: For the considering factors, some additional factors are also need to be consider, such as instrument calibration effects (spectral calibration and radiance calibration). In addition, the spectral information in specific pixel can be varied. For this reason, the spectral stability of hyperspectral imager has to be presented. 

3) What is the difference between reflectance and normalized radiance in Table 4?

4) If surface information does not match in the predefined category, how is the algorithm will be estimated?

5) In section 2, detailed classification algorithm has to be explained. 

6)Although the TP and FN value is suitable to the F1 score calculation, this statistical score is dissatisfied to explain the algorithm performance.

Reviewer 2 Report

This research proposes a deep learning method for annotating HSI automatically, and further classify spectral signals in areas of dense vegetation. This is a meaningful reserach topic, but several reasons make me decide to reject this manuscript as its current form. 

  1. There are something wrong in the pdf file. All hyperlinks/refences/figure number are missing, which makes the entire flow hard to follow.
  2. A high-level diagram is needed in the manuscript. Currently, it is little messy. 
  3. I think it is necessary to validate the performance of MaskCNN before doing pixel-level classification. Because, the protogenic MaskCNN is not design for single-channel grayscale images. There are some obvious misdetected objects in different channels.

Reviewer 3 Report

This paper proposed an automatic annotation of HSI with deep learning. The method is correct, and the authors have done a good work which is important to the field of HSI image analysis. I have some minor suggestions:

  1. The purpose of the proposed method should be more explicit. In the abstract, the aim is to detect people and cars, which are not mentioned in the title. In addition, when mentioned hyperspectral image we assume it is referred to as satellite hyperspectral image. But in this paper, it refers to the camera image. I think ‘camera image ’ should be highlighted in the title.
  2. In the abstract, it is said: “the current passive hyperspectral imaging technology still has some constraints”. The constraints should be explained more clearly in the abstract.
  3. The novelty of this paper should be highlighted. For instance, the authors can explain why it is difficult to detect people or other objects in dense vegetation areas?
  4. A flowchart of the method used should be added. In the materials and methods section, there are too many details, while the novelty and strength of the proposed method are not highlighted.  

Round 2

Reviewer 1 Report

This manuscript is well revised in the present version. 

Reviewer 2 Report

The modifications successfully addressed my previous concerns.